# Direct Current to Digital Converter (DIDC): A Current Sensor

**DOI:** 10.3390/s24216789

**Published:** 2024-10-22

**Authors:** Saeid Karimpour, Michael Sekyere, Isaac Bruce, Emmanuel Nti Darko, Degang Chen, Colin C. McAndrew, Doug Garrity, Xiankun Jin, Ilhan Hatirnaz, Chen He

**Affiliations:** 1Department of Electrical and Computer Engineering (ECpE), Iowa State University, Ames, IA 50011, USA; msekyere@iastate.edu (M.S.); ibruce@iastate.edu (I.B.); ntidarko@iastate.edu (E.N.D.); 2NXP Semiconductors, Chandler, AZ 85224, USA; colin.mcandrew@nxp.com (C.C.M.); doug.garrity@nxp.com (D.G.); robert.jin@nxp.com (X.J.); ilhan.hatirnaz@nxp.com (I.H.); chen.he@nxp.com (C.H.)

**Keywords:** reliability, measurement, ADC, DIDC, CMOS, VLSI, SAR

## Abstract

This paper introduces a systematic approach to the design of Direct Current-to-Digital Converter (DIDC) specifically engineered to overcome the limitations of traditional current measurement methodologies in System-on-Chip (SoC) designs. The proposed DIDC addresses critical challenges such as high power consumption, large area requirements, and the need for intermediate analog signals. By incorporating a current mirror in a cascode topology and managing the current across multiple binary-sized branches with the Successive Approximation Register (SAR) logic, the design achieves precise current measurement. A simple comparator, coupled with an isolation circuit, ensures accurate and reliable sensing. Fabricated using the TSMC 180 nm process, the DIDC achieves 8-bit precision without the need for nonlinearity calibration, showcasing remarkable energy efficiency with an energy per conversion of 1.52 pJ, power consumption of 117 µW, and a compact area of 0.016 mm². This innovative approach not only reduces power consumption and area, but also provides a scalable and efficient solution for next-generation semiconductor technologies. The ability to conduct online measurements during both standard operations and in-field conditions significantly enhances the performance and reliability of SoCs, making this DIDC a promising advancement in the field.

## 1. Introduction

The continuous advancement of semiconductor technology has positioned Integrated Circuits (ICs) as a key component in a wide array of applications, from everyday consumer electronics to critical systems in the automotive, aerospace, and medical industries [1,2,3,4,5]. These applications necessitate precise and effective real-time monitoring of IC operational parameters, including quiescent and operational currents. This monitoring is crucial as it allows designers to identify unusual chip behavior and assess their health, thereby proactively addressing potential issues that could affect system performance and reliability.

This approach not only enhances our comprehension and control of IC performance, but also plays a significant role in improving the reliability and safety of the systems that rely on it. For example, to improve the reliability of ICs, it is essential to mitigate the effects of aging phenomena such as hot carrier injection (HCI), bias temperature instability (BTI), time-dependent dielectric breakdown (TDDB) and electromigration (EM) [6,7,8,9,10]. Among these, EM in the Power Delivery Networks (PDNs) of ICs is a major concern, largely due to its dependence on current density and temperature. Precise current measurement is key to managing these effects. By calculating current density using the known cross-sectional area of the PDN and combining it with temperature data, the expected Median Time to Failure (MTTF) due to EM can be estimated. This estimation is performed using Black’s Equation, a widely accepted model in IC reliability studies [11].

Power consumption is another crucial aspect that directly influences the performance and lifespan of circuits. One element of power consumption that has received considerable focus is the quiescent current, which is the current a device draws when it is idle or in standby mode. A lower quiescent current is preferred because it means that less power is consumed during idle periods, thereby prolonging the overall battery life. Indeed, measuring the quiescent current is extremely important as it offers valuable insights into a device’s power efficiency during its idle state. This can aid in the early identification of potential problems, the optimization of power management strategies, and ultimately the extension of the device’s operational lifespan. As such, the precise and ongoing monitoring of quiescent current is a critical part of ensuring the optimal performance and longevity of ICs [12].

Traditional methods of current measurement, which are commonly used in SoC designs, involve the use of ADCs and thin-film resistors. These components work in tandem to convert the current into a voltage signal, which is then transformed into a digital value. However, this method presents certain inefficiencies, particularly when applied to SoC designs that incorporate multiple currents to be measured. The process of converting current to voltage and then to digital values introduces challenges such as higher power consumption and error accumulation. The presence of multiple current sources in SoC designs further exacerbates these challenges. Each current may require its own set of ADCs and thin-film resistors, leading to an increase in the system’s power consumption and physical footprint. Additionally, this can also increase the potential for errors [13,14]. Therefore, while traditional methods of current measurement are widely used, they present significant challenges when applied to complex SoC designs.

The direct conversion of current into digital values has emerged as a promising solution to the challenges faced in the field of ICs. This concept, known as Direct-to-Digital Converters (DxDCs), has been gaining significant attention in recent years [15,16,17]. DxDCs streamline the process of data acquisition by eliminating the need for intermediate analog signals, thereby enhancing the efficiency of the system.

In this context, we propose a novel method that directly measures the current and functions effectively as a current sensor. This ADC stands out by enabling online measurements both during normal SoC operations and in-field conditions. This capability ensures that SoCs can be monitored and managed effectively throughout their operational lifecycle, thereby enhancing both performance and reliability.

The remainder of this paper is organized as follows: Section 2 delineates the materials and methodologies employed in the design of the DIDC. This section is bifurcated: the first subsection elucidates the circuit design of the DIDC, encompassing its transistor-level realization, while the second subsection expounds on the design of the testbench utilized to conduct the Histogram test on the detailed design of the proposed DIDC. Section 4 presents the simulation results, thereby validating the design. Finally, Section 5 offers the conclusion.

## 2. Circuit Design Method

The proposed DIDC realization is illustrated in Figure 1. $I_{S}$ is the current that the DIDC generates corresponding to the digital value. A Banba band-gap reference [18] is presumed to be available in this figure to generate the biasing voltages. The binary weighted current generation branches are controlled by a SAR logic design. A simple comparator is used to decide the output voltage level to achieve the desired SAR logic search. Gate switching is employed in this design to control the current flowing into the binary-weighted branches.

### 2.1. Binary-Weighted Branches Design

Figure 2 is used to explain the design concept:

A current mirror is utilized in the cascode topology to minimize systematic errors caused by different $V_{DS}$ voltages at $M_{1}$, $MB_{D}$, and binary-weighted branches (e.g., $MB_{0-7}$). This arrangement is designed to generate binary-weighted currents, $I_{MB_{0}}$, …, $I_{MB_{7}}$ from the reference current, $I_{ref}$. The drain current, $I_{d}$, in each NMOS transistor can be expressed as given in Equation (Equation 1).

According to this equation, the drain current of an NMOS transistor is correlated with its size, electron mobility (μ), oxide capacitance per unit area ($C_{ox}$), a function of its $V_{GS}$ and $V_{BS}$, and a function of $V_{DS}$. According to Figure 2 and equation
(1)$$I_{d}=\frac{W}{L}\mu C_{ox}F_{1}\left(V_{GS},V_{BS}\right)F_{2}\left(V_{DS}\right),$$
the drain current of an NMOS transistor is correlated with its size, electron mobility (μ), oxide capacitance per unit area ($C_{ox}$), a function of its $V_{GS}$ and $V_{BS}$, and a function of $V_{DS}$. According to Figure 2, equation
(2)$$I_{ref}=\frac{W_{1}}{L_{1}}\mu C_{ox}F_{1}\left(V_{GS1},V_{BS1}\right)F_{2}\left(V_{DS1}\right),$$
and equation
(3)$$I_{MBi}=\frac{W_{MBi}}{L_{MBi}}\mu C_{ox}F_{1}\left(V_{GSMBi},V_{BSMBi}\right)F_{2}\left(V_{DSMBi}\right),$$
in which *i* is an integer from 0 to n−1 and n is the number of desired bits, since the gates and sources of $M_{1}$ and $M_{Bi}$ are tied together, it is implied that
(4)$$F_{1}\left(V_{GSMBi}\right)=F_{1}\left(V_{GS1},V_{BS1}\right)$$

Employing (Equation 4) and solving (Equation 2) in (Equation 3), (Equation 5) is derived as equation
(5)$$I_{MBi}=I_{ref}\frac{W_{MBi}L_{1}}{W_{1}L_{MBi}}\frac{F_{2}\left(V_{DSMBi}\right)}{F_{2}\left(V_{DS1}\right)}.$$

By maintaining the drain-source voltage $V_{DS}$ of the transistors at an equal level, it is possible to eliminate the $F_{2}$ term. This leads to the current flowing through binary-weighted branches being solely dependent on the ratio between the current reference generator and the base transistor of the respective binary-weighted branch as shown in equation
(6)$$I_{MBi}=I_{ref}\frac{W_{MBi}L_{1}}{W_{1}L_{MBi}}$$

When the SAR logic initiates a search for the $I_{S}$ value, each branch establishes a connection to the midpoint from the most significant bit (MSB) to the least significant bit (LSB), allowing the binary-weighted current to traverse the branch. As per equation
(7)$$V_{o}=\left\{\begin{array}{cc} V_{DD}, & \mathrm{if}\;I_{DAC}<I_{S}\to R=1\\ V_{CM}, & \mathrm{if}\;I_{DAC}=I_{S}\\ 0, & \mathrm{if}\;I_{DAC}>I_{S}\to R=0 \end{array}\right.,$$

The value $R_{i}$ is generated, digitally representing the participation of the corresponding branch in the total current. $V_{CM}$ here refers to the common node voltage, and in this case it is $\frac{V_{DD}}{2}$. In a balanced state, the $I_{S}$ value is expressed as equation
(8)$$I_{S}=I_{B_{D}}+CI_{ref},C=\sum_{i=0}^{n-1}R_{i}\frac{W_{MBi}L_{1}}{W_{1}L_{MBi}}$$

The design was executed in such a way that it remains stable in saturation once the DIDC reaches a balanced state.

In summary, the following steps can be followed to achieve a systematic design of the DIDC:**Choose the Maximum Current Range**: Define the maximum current range that the DIDC measures. This sets the full-scale measurement range of the converter.**Assign MSB Current**: Ensure that the current represented by the MSB is half of the total range. This means that the MSB current is half of the maximum measurable current, and the remaining bits are scaled accordingly.**Binary-Weight the Remaining Bits**: After setting the MSB current, continue with binary-weighted scaling for the remaining bits. Each subsequent bit represents half the current of the preceding bit, following a binary progression from the MSB to the LSB.**Set the Reference Current $I_{\mathrm{ref}}$**: Choose the reference current such that the MSB current is a multiple of $I_{\mathrm{ref}}$, typically $I_{\mathrm{ref}}=\frac{I_{\mathrm{MSB}}}{8}$.**Size the Base Transistor**: Assuming the reference current is supplied from a voltage source, the base transistor of the current reference can be sized using Equation (Equation 1). Ensure that the transistor operates with a gate-source voltage of $V_{GS}=\frac{V_{DD}}{2}$, where $V_{DD}$ is the supply voltage. This guarantees that the transistor remains in saturation, providing a stable current for the desired reference value.**Size the Cascode Transistors**: Size the cascode transistors to maintain the base transistors “comfortably in saturation.”**Size the Binary-Weighted Transistors**: Scale the transistors for each binary-weighted current branch according to their relative weight using Equation (Equation 6).**Verify SAR Logic and Measurement**: Ensure proper operation of the SAR logic to perform a binary search and generate the correct digital output for the measured current.

### 2.2. Comparator

A simple comparator design, consisting of two inverters in series, is used to generate a digital output based on the current value. The design depends on two key factors: the inverter threshold and the comparator resolution. The selection of an appropriate value for the comparator gain depends on the resolution of the DIDC. For example, to achieve clear logic for a DIDC with n-bit accuracy, it is assumed that the lower margin of the comparator is $V_{L}=V_{SS}+0.1$ and the higher margin is $V_{H}=V_{DD}+0.1$, necessitating a resolution of
(9)$$resolution=\frac{V_{H}-V_{L}}{2^{n}},$$These two voltages depend on technology in which design is implemented. Given that the node driving the comparator is a high-impedance node, employing this simple comparator does not introduce errors in the measurement due to noise.

### 2.3. Isolation Circuit

To maintain circuit balance and ensure precise current sensing, an isolation circuit is utilized. When using a PMOS current mirror as the source of $I_{S}$, it is essential to isolate the sensing node from the primary node to maintain a constant drain-source voltage $V_{DS}$ for the reference and mirrored currents. The PMOS transistor fulfills this role, as illustrated in Figure 1. To prevent the sensing current from being influenced by varying voltages in the main node due to different currents, a negative feedback loop is established using a folded cascode operational amplifier with an NMOS input pair, capable of operating at voltages above $V_{DD}$.
(10)$$V_{g}=V_{gs_{PMOS}}+V_{LDO\_OUT}$$

## 3. Fabrication and Measurement

The measurement process is critical in verifying the operational principles of the 8-bit DIDC. This section outlines the measurement methodology, including the test setup, signal generation, and the evaluation of performance metrics such as accuracy, power consumption, and INL/DNL. Measurements are conducted using a programmable current source in conjunction with an FPGA, and the results are validated using both ramp testing and spectrum analysis. In this example, a maximum current of 200 µA is measured, with a focus on ensuring that the DIDC maintains its 8-bit resolution across varying operating conditions. This value is chosen according to the requirement of an NXP LDO designed using the current matching technique introduced in [19]. In this application, the LDO has an output current ranging from 100 μA to 20 mA, and this current is sampled to a range of 1 μA to 200 μA. The reference current chosen is $I_{ref}$ = 12.5 μA. This reference current should be chosen to generate biasing voltages for the base NMOS transistors in a way that ensures that they are "comfortably saturated" without requiring large transistor sizes.

Additionally, the MSB branch has a current of 8 $I_{ref}$. For each subsequent branch from the MSB to the LSB, the current is halved. In other words, the second MSB bit has a weight of 4 $I_{ref}$, while the LSB bit has a weight of 1/16 $I_{ref}$. The dummy branch has the same value of 1/16 $I_{ref}$. This concept is explained in the Section 4 in detail.

The layout of the different blocks are shown in Figure 3:

Table 1 is employed to reveal the size of the binary-weighted array used in this example.

The microscopic view of the manufactured DIDC is shown in Figure 4, including the presence of blocks related to other projects that are not the focus of this article.

It appears that dummy fill is used extensively. However, dummy blockers can be strategically placed to avoid filling certain areas while still meeting density requirements. Each metal and poly layer has a “dm” layer that can be used to prevent the script from inserting dummies. Although removing the dummy fill may affect the consistency and yield of the etching, it should only be applied to nonsensitive areas. Removing the dummy fill can help reduce some parasitics, depending on the purpose of the circuit.

The current to be measured, $I_{S}$, is generated using a programmable two-stage high-side current source from Texas Instruments [20] and shown in Figure 5. This current source is controlled via a DAC, allowing for precise voltage regulation and current generation. The generated current is fed into the DIDC circuit, which converts the analog current into a digital output through its SAR logic. The output is then sent to an FPGA, which handles data acquisition and processing. To ensure accuracy, a folded cascode operational amplifier is employed in the isolation circuit to maintain a stable voltage reference across the current mirror branches.

A critical consideration in the design of this circuit is the response characteristics of the NMOS and PMOS transistors employed. These transistors possess large input capacitances, which reduce their response time. Additionally, the node connecting the circuit to the DIDC exhibits high impedance, potentially limiting the final operational speed of the DIDC. A compensation circuit is also implemented to prevent instability due to the feedback loop and to avoid output current oscillation. The current is controlled using the DAC80508 from TI, as described by the following equation:(11)$$I_{S}=\frac{\mathrm{code}}{2^{k}}\;\cdot \;\frac{R_{2}}{R_{1}\;\cdot \;R_{3}}\;\cdot \;V_{\mathrm{ref}}$$

In this setup, parameter k denotes the accuracy of the DAC, which can be configured to 12, 14, or 16 bits. The reference voltage $V_{ref}$ is set to 1.25 V using an external resistor divider network. Resistors $R_{1}$ to $R_{3}$ are employed to adjust the current and to cancel the DAC output voltage offset.

Voltage sources are required from an on-chip voltage reference to initiate the chip. However, as the design is standalone, TI LDOs, including LP2981I, TPS7A0212, TPS7A0215, TLV75509, TPS7A0212, and TPS7A0218, are utilized. The purpose of these LDOs are as follows, respectively: 3.6 V is used to supply the DAC and Op Amp circuit with the necessary voltages, 1.2 V is employed for generating $V_{ref}$, 1.5 V functions to set the reference voltage of the isolation circuit (which can be adjusted as required), 0.9 V is employed as the bias voltage $V_{bias}$, and 1.8 V acts as the analog supply voltage $AV_{DD}$ for the chip. All of these LDOs are supplied by a voltage generator that is set to 5 V. This entire configuration is mounted on a printed circuit board (PCB) shown in Figure 6a. The circuit’s performance is evaluated using the DE2-115 FPGA from TI. To initiate and adjust the DAC code, the Serial Peripheral Interface (SPI) standard is required for communication with the DAC. The core of the SPI is implemented in Verilog and connected to JP4 on the FPGA. Additionally, SAR logic is implemented to control the binary-weighted array, with different voltage logics. The SAR is implemented on FPGA pin JP5 with a voltage of 1.8 V. The Universal Asynchronous Receiver–Transmitter (UART) is implemented on the FPGA to transmit data from the FPGA to a computer. An analog device logic analyzer ensures correct communication between the DAC and the FPGA.

The operation of the DIDC was evaluated across a range of temperatures. To maintain the circuit at the desired temperature, a test equity half-cubic temperature chamber was utilized. To ensure that the temperature remained at the desired level, a temperature sensor was used. The setup of this experiment can be seen in Figure 6c,d.

## 4. Results

In this section, we present the results of the proposed DIDC design as outlined in the previous sections. Our evaluation encompasses both simulation and experimental data to validate the performance and functionality of the DIDC under various conditions. The analysis includes key metrics such as accuracy, power consumption, and area efficiency, as well as the comparison of theoretical predictions with actual measured outcomes.

The performance of the DIDC was evaluated through a series of experiments, utilizing an 8-bit resolution as implemented with the TSMC 180 nm process. We first discuss the simulation results, which provide an initial validation of the design parameters and functionality. Subsequently, we present the experimental measurements obtained from the fabricated chip, highlighting the operational efficacy of DIDC in practical scenarios.

### 4.1. Simulations

#### 4.1.1. Process, Local Mismatch, and Temperature Sensitivity

To evaluate the sensitivity of the design to process and temperature variations, Monte Carlo simulations are conducted at three temperature corners: −50 °C, 27 °C, and 150 °C, covering the AEC-Q100 Grade 0 temperature range. This approach is necessary for understanding how variations in transistor parameters affect circuit performance. By statistically analyzing thousands of possible variations at each temperature corner, the simulation helps predict and quantify deviations in key performance metrics of the DIDC, including the current deviation of each individual binary branch from its nominal value.

The evaluation process involves fixing the input node of the comparator at a meta-stable position and activating each individual binary-weighted branch. The currents flowing through these branches are adjusted according to their nominal values, with deviations caused by temperature and mismatches accounted for using a voltage-controlled current source (VCCS).

To assess this statistical behavior, a useful parameter is standard deviation. Standard deviation helps in understanding the shape of the data distribution. For example, in a normal distribution (bell-shaped curve), approximately 68% of the data falls within one standard deviation of the mean, about 95% falls within two standard deviations, and approximately 99.7% falls within three standard deviations. Monte Carlo simulations typically assume a normal distribution, but deviations from these percentages can indicate non-normal or skewed distributions:(12)$$\sigma =\sqrt{\frac{1}{N}\sum_{j=1}^{N}{\left(x_{j}-\mu \right)}^{2}}$$In this equation, σ represents the standard deviation, *N* is the number of data points, $x_{j}$ denotes each individual data point, and μ is the mean (average) of the data points. By evaluating the current in each branch and calculating the mean and standard deviation (STD), we can assess the effects of process and temperature variations on the circuit’s performance.

This assessment is vital because the circuit operates under the principle of superposition. If the deviations of each individual binary branch remain within specified limits, it ensures that the total error across the entire circuit is also within acceptable bounds. The superposition principle means that the total error is the sum of the errors from each component, so managing the error of each branch helps keep the overall error within manageable levels. If the total error due to process and temperature variations is less than 1 LSB (Least Significant Bit) in the current, then the DIDC’s accuracy is at least as precise as the resolution represented by the LSB.

Furthermore, errors propagate from the MSB to the LSB in a predictable manner. If the MSB error is less than 0.5 LSB, the error in the next MSB bit (second most significant bit) is less than 0.25 LSB, and this trend continues, with each subsequent bit’s error being half of the previous bit’s error. Therefore, if the MSB error is within 0.5 LSB, the overall system error, including all less significant bits, remains well within the 1 LSB limit. This ensures that the accuracy of the system aligns with the precision level indicated by the LSB.

As specified above, this design covers an application range of 1 mA to 20 mA. The total current, determined by downscaling the maximum current $I_{FS}$, is 200 uA. In an n=8-bit DIDC, the current for each bit is calculated using Equation (Equation 14),
(13)$$I_{MBi}=\frac{I_{FS}}{2^{n-i}},$$
and
(14)$$ratio_{MBi}=\frac{I_{MBi}}{I_{ref}},$$
where i=0 to 7, and ${\mathrm{ratio}}_{MBi}$ represents the mirroring index of the $i^{\mathrm{th}}$ branch. For example, at i=7, $I_{MB7}$ is 100μA with ${\mathrm{ratio}}_{MB7}=8$, while the current decreases to $I_{MB0}=781.35\;\mathrm{nA}$ with ${\mathrm{ratio}}_{MB0}=1/16$ in the smallest branch. These values indicate the nominal current for each bit.

The results of a Monte Carlo simulation with 1000 points at three different temperatures for the three MSB branches, simulated in Specter, are shown in Figure 7.

As stated before, to ensure that the error due to process and temperature variations remains minimal, the deviation of the MSB bit current should be less than half the LSB current. Here, as previously specified, $I_{MB0}=I_{LSB}=781.35\;\mathrm{nA}$. To achieve 8-bit accuracy, the variation in the MSB bit must satisfy
(15)$$I_{MB7_{\mathrm{mean}}}-\frac{I_{LSB}}{2}\leq I_{MB7_{\mathrm{MC}}}\leq I_{MB7_{\mathrm{mean}}}+\frac{I_{LSB}}{2}.$$

In this example, as shown in Figure 7, the worst behavior in terms of mismatch occurs at the coldest temperature. The STD of the MSB current, depicted in Figure 7c, is evaluated to be $\sigma_{MB7}=138\;\mathrm{nA}$, and $3\sigma_{MB7}\approx I_{LSB}/2$. Following this trend to the lower bits, $3\sigma_{MB6}\approx I_{LSB}/4$ and $3\sigma_{MB5}\approx I_{LSB}/8$. This indicates that as we move to lower bits, the standard deviation decreases proportionally, maintaining the required accuracy for an 8-bit design.

The improvement in performance observed at higher temperatures, as shown in Figure 7, can be attributed to the temperature dependence of the transistor characteristics, specifically the $V_{th}$ and μ in MOSFET devices.

At colder temperatures, transistor mismatch becomes more pronounced due to increased variation in $V_{th}$ and reduced carrier mobility, leading to worse performance in terms of mismatch. As temperature increases, the variation in $V_{th}$ tends to decrease, and mobility improves, reducing the effect of mismatch across the transistors in the current mirror array. This results in better matching between the transistors and thus improved linearity and performance.

Table 2 shows the complete values for every individual bit in terms of STD and mean.

The observed variations are closely associated with the dimensions of the base transistors. According to Equation (Equation 16),
(16)$$\sigma =\frac{A}{\sqrt{WL}}$$
where *W* and *L* represent the width and length of the transistor, respectively, and *A* is a constant specific to a particular parameter, such as current, provided in the Process Design Kit (PDK) [21], as the dimensions of the transistor increase, the mismatch decreases, thereby improving the overall performance and reliability of the circuit.

#### 4.1.2. Transient

To accurately assess the performance of an Analog-to-Digital Converter (ADC), it is essential to examine two critical parameters: Integral Non-Linearity (INL) and Differential Non-Linearity (DNL). These parameters can be effectively evaluated using a ramp test.

**Integral Non-Linearity (INL)** quantifies the deviation of the actual ADC transfer function from an ideal straight line that represents the expected linear response. During a ramp test, a linearly increasing or decreasing input signal is applied to the ADC, and the output is recorded. INL at a given code *k* is defined as
(17)$$\mathrm{INL}\left(k\right)=V_{\mathrm{actual}}\left(k\right)-V_{\mathrm{ideal}}\left(k\right),$$
where $V_{\mathrm{actual}}\left(k\right)$ is the actual output voltage corresponding to the digital code *k*, and $V_{\mathrm{ideal}}\left(k\right)$ is the ideal output voltage for that code. A high INL value indicates significant deviations from linearity, leading to errors in the conversion process. As a result, INL directly impacts the overall accuracy of the ADC by introducing systematic errors throughout the input range.

**Differential Non-Linearity (DNL)** measures the deviation between the actual and ideal step sizes between adjacent digital codes. The DNL at a specific code *k* is expressed as
(18)$$\mathrm{DNL}\left(k\right)=\frac{\Delta V_{\mathrm{actual}}\left(k\right)}{\Delta V_{\mathrm{ideal}}}-1,$$
where $\Delta V_{\mathrm{actual}}\left(k\right)$ represents the actual step size between codes *k* and k+1, and $\Delta V_{\mathrm{ideal}}$ is the ideal step size. In a ramp test, by carefully analyzing these step sizes as the input voltage increases or decreases, one can calculate the DNL for each code. High DNL values can result in missing codes, where certain input levels do not correspond to any digital code, potentially leading to data loss. Therefore, DNL primarily affects the precision of the ADC by ensuring the consistency and uniformity of the conversion steps.

**Relationship and Accuracy Implications**: The ramp test is instrumental in identifying and quantifying INL and DNL, which are closely related to the ADC’s accuracy and precision. INL affects the overall accuracy by indicating how much the ADC’s output deviates from an ideal linear response, with lower INL values reflecting higher accuracy. DNL affects the ADC’s precision by ensuring that each step in the input voltage is consistently represented in the digital output. Minimizing both INL and DNL is crucial for achieving reliable, accurate, and precise analog-to-digital conversion.

These points should be considered to understand the INL and DNL curves:


**Evaluate INL:**
**Check for Maximum Deviation:** Identify the points on the INL curve where the deviation is largest (positive or negative), showing the worst-case non-linearity.**Linearity:** Ideally, the INL curve should be as flat as possible. A flat INL indicates good linearity of the ADC across its entire input range.**Systematic Errors:** Look for systematic trends in the INL curve (e.g., consistently increasing or decreasing), which may indicate issues in the ADC design or external influences like power supply variations.



**Evaluate DNL:**
**Uniformity of Steps:** Analyze how close the DNL values are to zero. A DNL of zero means the step sizes are uniform, which is ideal.**Check for Missing Codes:** If DNL <−1 at any point, it indicates a missing code, meaning some input ranges are not represented in the output.**Random Noise vs. Systematic Errors:** Random variations in DNL might indicate noise, whereas consistent patterns could suggest systematic errors in the ADC design.



**Interpret Results:**
**Good ADC Performance:** A good ADC has INL and DNL close to zero across the entire input range, indicating accurate and precise conversion.**Poor ADC Performance:** Large deviations in INL or DNL, especially in certain regions, suggest potential problems in the ADC’s design or implementation, leading to inaccuracies.


In this example, the input of the DIDC was transferred from 0 μA to 205 μA, in the worst corner of the Monte Carlo simulation, that is, the corner with the largest mismatch in three different temperatures. One consideration for testing the performance of the DIDC is the clock speed of the SAR logic. According to Figure 1, the input node of the comparator is a high impedance node and has a large parasitic capacitor $C_{p}$. To ensure that the conversion is performed correctly, the voltage of this node should settle according to the capacitor voltage Equation (Equation 19),
(19)$$V_{comp}=C_{p}\frac{dI}{dt},$$
where
(20)$$I=I_{S}-I_{DAC}.$$

As *I* has a higher value, it changes more quickly with time, resulting in faster voltage settling. However, with smaller currents, it takes longer to settle. To avoid any missing codes, this should be carefully considered. The internal dynamics of the circuit during conversion can be illustrated in Figure 8.

Considering this in the conversion, the INL and DNL plot in $f_{clk}$ = 125 kHz is shown in Figure 9 according to the post-layout simulation results.

The DNL plot demonstrates the step size variation between adjacent digital codes, while the INL plot highlights the overall linearity of the converter across the full input range. These results show that the DIDC design maintains high precision and linearity, demonstrating good robustness across the specified temperature range and clock speed, ensuring reliable performance under varying conditions.

### 4.2. Measurement

#### 4.2.1. Ramp Test

As stated in the Simulation section, the ramp test is employed to evaluate the INL and DNL of the DIDC. During this test, a linearly increasing current is generated by the DAC and applied to the input of the DIDC. The SAR logic within the DIDC quantizes the input current into corresponding digital codes. The INL is calculated by comparing the actual digital output with the expected linear response, while the DNL is assessed by measuring the consistency of the step sizes between adjacent digital codes. These metrics provide insight into the overall accuracy and linearity of the current-to-digital conversion process. To perform this test, the test bench shown in Figure 7 is used, where a gradual voltage change is applied to achieve a gradual current change with 14-bit accuracy, resulting in 64 hits.

This evaluation is conducted at different clock speeds and temperatures, specifically at −10 °C and 100 °C in a temperature chamber, and at room temperature, as shown in Figure 6b–d. The reason for not testing at higher and lower temperatures stems from two limitations: the components on the PCB are designed to operate between −25 °C and 125 °C, and the temperature chamber operates between −40 °C and 130 °C.

Another option to control the temperature is to use a micro-bath, but the oil used in a micro-bath is prohibitively expensive and begins to evaporate at 125 ° C. Since our goal was to measure performance under hot and cold conditions rather than precise temperature control, we ensured that the temperature of the die was sufficiently hot and cold for this purpose.

The state machine governing the measurement process during the ramp test is illustrated in Figure 10.

The conversion results at different clock speeds are presented as INL and DNL plots in Figure 11.

Based on the analysis of the test bench, we determined that to maintain the desired accuracy, the system should operate at a clock speed of 25 kHz. This clock speed ensures that the deviations remain within acceptable limits, specifically under ±0.7 LSB, as shown in the previous figures.

To further evaluate the performance under different temperature conditions, we conducted tests at −10 °C and 100 °C. The results at these temperatures were also plotted for INL and DNL using the 25 kHz clock speed, as this configuration provided the most reliable accuracy in our initial testing. The following figures present the INL and DNL plots for these extreme temperature conditions, ensuring that the system maintains its accuracy in a wide range of operating environments; Figure 12.

From this evaluation, the INL and DNL curves at a clock speed of 25 kHz and room temperature indicate that the system achieves an accuracy within 1 LSB, suggesting that 8-bit accuracy is maintained under these conditions. However, as the clock speed increases, the accuracy diminishes. This reduction in precision is likely due to the capacitive effects in the circuit, as depicted in Figure 7.

Another trend verified in the measurements, which was also predicted by the simulations, is that the sensor exhibits better performance at higher temperatures compared to colder temperatures. This can be concluded by comparing the DNL curves at different temperatures: at −10 °C, the DNL deviation is around ±0.8 LSB, at room temperature it decreases to ±0.7 LSB, and at 100 °C it improves further to ±0.5 LSB.

#### 4.2.2. Spectrum Test

In addition to static tests like the ramp test, the dynamic performance of the DIDC is evaluated using spectrum analysis. A sine wave input current is generated using MATLAB-2024a, Matworks, Torrance, CA, USA which is then converted into corresponding DAC codes. These codes are fed into the DIDC, and the digital output is analyzed in the frequency domain using Fast Fourier Transform (FFT). This analysis provides key metrics such as Signal-to-Noise Ratio (SNR), Spurious-Free Dynamic Range (SFDR), and Total Harmonic Distortion (THD). These metrics help assess the converter’s ability to accurately represent high-frequency signals and its susceptibility to noise and distortion.

To evaluate the dynamic performance of the DIDC, a spectrum analysis was performed using a coherent sampling approach. Coherent sampling is a technique used in ADC testing where the input sine wave frequency is chosen so that an integer number of sine wave cycles fits exactly into the time window of the data acquisition. This ensures that the sampled data do not have any discontinuities at the beginning and end of the acquisition window, which could cause spectral leakage and distort the frequency spectrum.

The relationship between the parameters for coherent sampling is given by
(21)$$J=M\;\cdot \;\frac{f_{\mathrm{sig}}}{f_{\mathrm{sample}}}$$
where *J* is the number of sine wave cycles within the sampling window, *M* is the number of samples, $f_{\mathrm{sig}}$ is the frequency of the input signal, and $f_{\mathrm{sample}}$ is the sampling frequency.

The sampling frequency $f_{\mathrm{sample}}$ is determined by the following equation:(22)$$f_{\mathrm{sample}}=M\;\cdot \;\frac{f_{\mathrm{clk}}}{2^{n}}$$
where $f_{\mathrm{clk}}$ is the clock frequency and *n* represents the resolution (in bits) of the DIDC.

#### Coherent Sampling Setup

The following steps were taken to set up the coherent sampling:

**Input Signal Generation**: A sine wave signal was generated in MATLAB with a peak-to-peak amplitude slightly lower than the maximum current that the DIDC could handle. The frequency of the sine wave was selected so that an integer number of cycles fit exactly within the data acquisition time window. This ensured that the sampled data did not contain any discontinuities at the beginning or end of the acquisition window, thereby preventing spectral leakage.

**Conversion to DAC Codes**: The generated sine wave current signal was converted into corresponding DAC input codes using Equation (Equation 11). These codes were then programmed into the DAC via the FPGA.

#### Spectrum Analysis Procedure

**Data Acquisition**: The output of the DIDC, after the sine wave signal was input, was collected, and preprocessing was carried out to remove any potential noise or artifacts from the data. The mean (average) of the signal was calculated as follows:(23)$$\mu_{x}=\frac{1}{N}\sum_{n=0}^{N-1}x\left[n\right]$$
where $\mu_{x}$ is the mean of the signal, *N* is the total number of samples, and x[n] represents the *n*th sample of the signal.

The signal was then adjusted by subtracting the calculated mean:(24)$$x_{\mathrm{processed}}\left[n\right]=x\left[n\right]-\mu_{x}$$
where $x_{\mathrm{processed}}\left[n\right]$ is the mean-adjusted (processed) signal.

**FFT and Magnitude Spectrum**: The preprocessed data, now centered around zero, were transformed from the time domain to the frequency domain using the Fast Fourier Transform (FFT),
(25)$$X\left[k\right]=\sum_{n=0}^{N-1}x_{\mathrm{processed}}\left[n\right]e^{-\frac{j2\pi kn}{N}}$$
where X[k] is the *k*th frequency component in the spectrum and $e^{-\frac{j2\pi kn}{N}}$ represents the complex exponential component of the FFT.

From the resulting frequency spectrum, key parameters were calculated. The fundamental power, which represents the main signal component, was determined as
(26)$$P_{\mathrm{fundamental}}={\left|X\left[k_{\mathrm{fund}}\right]\right|}^{2}$$
where $P_{\mathrm{fundamental}}$ is the power of the fundamental frequency component, $k_{\mathrm{fund}}$ corresponds to the index of the fundamental frequency, and $|X\left[k_{\mathrm{fund}}\right]{|}^{2}$ represents the squared magnitude of the fundamental frequency component.

The noise power, excluding the fundamental component, was calculated by summing the remaining power in the spectrum:(27)$$P_{\mathrm{noise}}=\sum_{k\neq k_{\mathrm{fund}}}{\left|X\left[k\right]\right|}^{2}-P_{\mathrm{fundamental}}$$
where $P_{\mathrm{noise}}$ is the total noise power, $\sum_{k\neq k_{\mathrm{fund}}}{\left|X\left[k\right]\right|}^{2}$ represents the total power of all frequency components excluding the fundamental frequency.

Using these values, the Signal-to-Noise Ratio (SNR) was calculated:(28)$$\mathrm{SNR}=10\log_{10}\left(\frac{P_{\mathrm{fundamental}}}{P_{\mathrm{noise}}}\right)\;\mathrm{dB}$$

**Spurious-Free Dynamic Range (SFDR)**: The SFDR was determined as follows:(29)$$\mathrm{SFDR}=10\log_{10}\left(\frac{P_{\mathrm{fundamental}}}{P_{\mathrm{noise}}+\mathrm{distortion}}\right)\;\mathrm{dB}$$
where distortion represents the power of the most significant spurious tone or distortion in the signal.

**Effective Number of Bits (ENOB):** Finally, the Effective Number of Bits (ENOB) was derived from the SNR:(30)$$\mathrm{ENOB}=\frac{\mathrm{SNR}-1.76}{6.02}\;\mathrm{bits}$$

**Analysis of Results**: The spectrum analysis provided insights into the DIDC’s ability to accurately convert the input signal without introducing significant noise or distortion. The results, characterized by a clean spectrum with minimal artifacts, confirmed that the DIDC performed as expected under the test conditions.

This systematic approach to spectrum analysis allowed for a thorough evaluation of the DIDC’s dynamic performance, ensuring that it meets the required specifications for accurate and reliable current-to-digital conversion.

The result of the processing measured data is shown in Figure 13:

The ENOB of 7.21 bits reflects the resolution of the DIDC converter, taking into account both noise and distortion. An ENOB of 7.21 bits suggests that the DIDC operates close to its theoretical maximum resolution of 8 bits, with minimal degradation due to noise and distortion.

### 4.3. Comparison and Discussion

#### 4.3.1. DxDCs Comparision

Although the converters compared in Table 3 serve different purposes—ref. [15] being a Light-Dose-To-Digital Converter and [16] a temperature-to-digital converter—the type and application of these converters differ, and the comparison focuses on common critical parameters such as power consumption, area and energy efficiency, which are relevant across various DxDC designs.

This work demonstrates an energy per conversion of 1.52 pJ and power consumption of 117 µW. In comparison, ref. [16], despite its different application, exhibits an energy per conversion of about 9 nJ and power consumption of 2250 µW. This highlights the significant energy efficiency of the proposed design. Additionally, while [15] operates in a smaller process node (65 nm vs. 180 nm), our design achieves a significantly smaller area (0.016 mm² vs. 1.71 mm²). These metrics underscore the efficiency and compactness of our approach, making it particularly suitable for low-power and space-constrained applications, despite differences in their functional domains.

#### 4.3.2. Area

The total area occupied by the DIDC, as shown in the layout in Figure 3, is 0.016 mm^2^ in a 180 nm technology. The primary goal of this design is to eliminate the need for standalone ADCs and thick film resistors to convert current to voltage and then to a digital signal. For comparison, an ultra-small sigma-delta ADC introduced by Taylor [22] occupies 0.07 mm^2^ in a 65 nm technology, demonstrating a 4.375 times improvement in area efficiency, even before considering the scaling factor between the technologies.

Traditional designs often require thin-film resistors to convert current to voltage. For instance, a typical thin-film resistor, such as a 10 kΩ resistor in a 65 nm technology, can occupy approximately 0.04 mm² [23]. In contrast, our design achieves the same functionality without the need for such resistors, thus saving valuable area on the chip.

Moreover, in a SoC environment, where multiple instances of this structure may be required, replicating our design further reduces the overall area consumption compared to traditional approaches that necessitate additional resistors and ADCs. This highlights the scalability and efficiency of our approach, particularly in large-scale integrated systems.

#### 4.3.3. Power Consumption

The total power consumption of the chip in the worst case scenario was evaluated to be 0.144 mW. This figure encompasses all the power drawn from the DC voltage source within the circuit. Notably, approximately two-thirds of this power consumption originates from the isolation Op Amp, which was fabricated without optimization. It is estimated that by using an Op Amp specifically designed for this structure, the power consumption could be reduced by 50%.

In comparison, Taylor’s ADC [22] demonstrates a power consumption of 8 mW in its most power-efficient configuration. Even with the current non-optimized design, our approach offers a significant reduction, with a 55.5-fold decrease in power consumption. This dramatic reduction highlights the efficiency of our design, especially in applications within System-on-Chip (SoC) environments where precise current measurement is critical. This efficiency not only conserves energy but also minimizes heat generation, which is essential for maintaining the reliability and longevity of the SoC components.

## 5. Conclusions

In this paper, we introduced a systematic approach to designing a Direct Current-to-Digital Converter (DIDC), exemplified through an 8-bit implementation aimed at advanced current measurement in System-on-Chip (SoC) applications. The proposed DIDC leverages a current mirror in a cascode topology, integrated with Successive Approximation Register (SAR) logic, to achieve precise 8-bit resolution without the need for nonlinearity calibration. Fabricated using the TSMC 180 nm process, this design showcases exceptional performance, including an energy per conversion of 1.52 pJ, power consumption of 117 µW, and a compact total area of just 0.016 mm^2^.

The results from both simulations and experimental evaluations validate the effectiveness of this design across various operating conditions, including extreme temperatures, confirming that the DIDC maintains accuracy within 1 LSB at different clock speeds and environmental conditions. The comparison with other DxDCs highlights the superior energy efficiency and compactness of our design, making it particularly suitable for low-power, space-constrained applications.

This work addresses the challenges of traditional current measurement methods by providing a scalable and efficient solution that eliminates the need for standalone ADCs and thick-film resistors. The DIDC’s capability for online measurements during both standard operations and in-field conditions significantly enhances the performance and reliability of SoCs, setting a new benchmark for future developments in semiconductor technologies.

Overall, the introduction of this systematic approach to DIDC design, illustrated through the 8-bit example, marks a significant advancement in current measurement technologies, offering a robust and reliable solution for modern SoC designs. Its integration into various SoC platforms has the potential to extend operational lifespans and improve the dependability of integrated circuits across diverse applications.

## Figures and Tables

**Figure 1 sensors-24-06789-f001:**
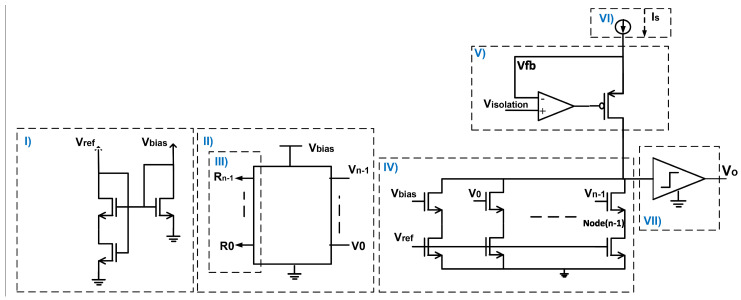
Conceptual overview of the DIDC design, divided into seven key sections: (**I**) Biasing circuit and reference current generation using a Banba band-gap voltage reference [18], (**II**) Switch array controlling gate-switching of cascade transistors via NOT gates, (**III**) SAR logic for controlling the switch array, (**IV**) Binary-weighted array for reflecting the reference current, (**V**) Isolation circuit separating the main and active nodes, (**VI**) Current $I_{S}$ generation within the SOC for DIDC measurement, and (**VII**) Comparator composed of two series-connected inverters.

**Figure 2 sensors-24-06789-f002:**
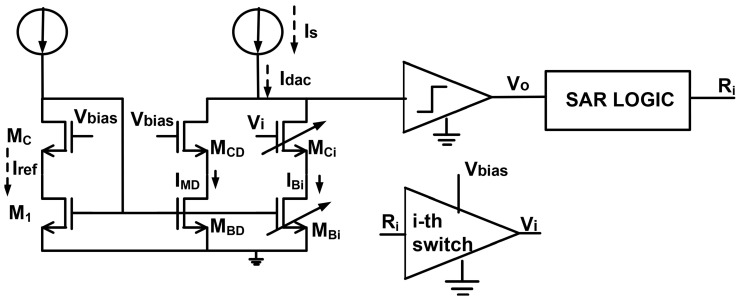
Schematic of the DIDC operation. The size of the base transistor, $M_{1}$, regulates the voltage $V_{ref}$ and the current, $I_{ref}$, and must be “comfortably in saturation”, implying $V_{DS}>3$ to $4\times V_{DS_{\mathrm{sat}}}$. The mechanism aims to solve for $I_{S}$, and at the conclusion of the conversion, it is expected to achieve $I_{S}=I_{dac}$, with the structure settling into a meta-stable balance at $V_{CM}$.

**Figure 3 sensors-24-06789-f003:**
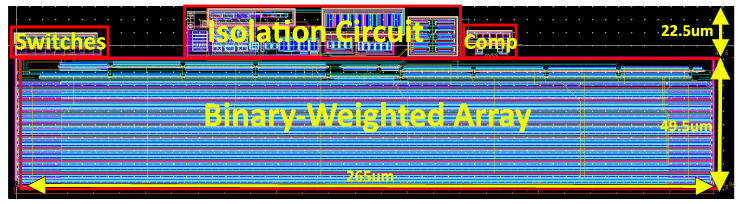
The layout of the fabricated DIDC. The locations of different blocks, including the binary-weighted array, switches, isolation circuit, and comparator, are indicated. As it can be seen, the main block is occupied by the binary-weighted array.

**Figure 4 sensors-24-06789-f004:**
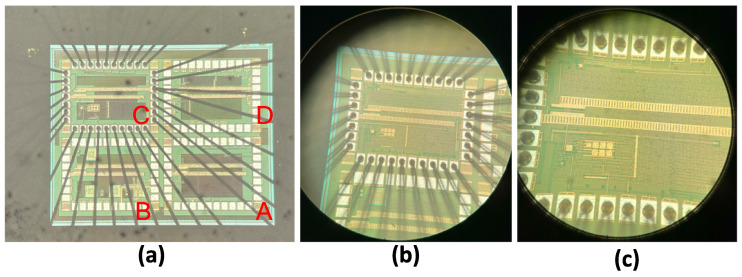
Microscopic view of the die: (**a**) As can be seen, there are four quadrants on the die, and only quadrant C is bonded, (**b**,**c**) shows the scaled view of quadrant C, in which DIDC is fabricated.

**Figure 5 sensors-24-06789-f005:**
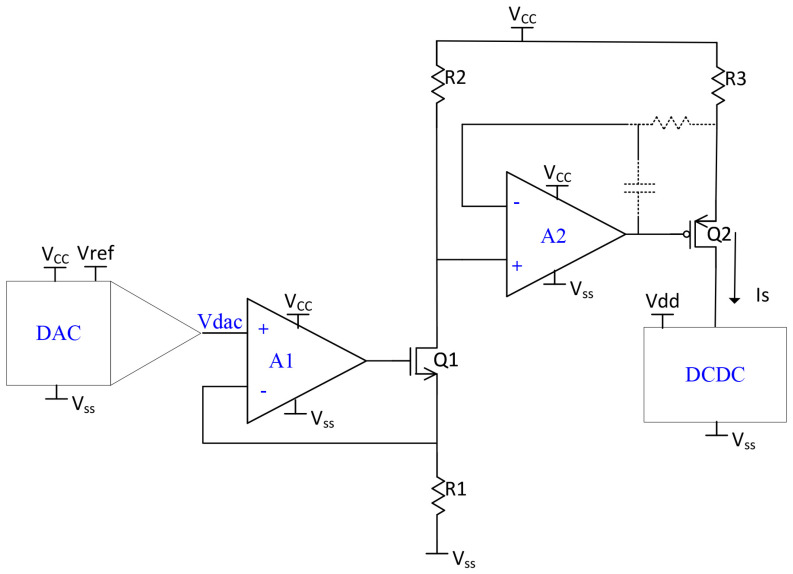
This schematic diagram represents a current source circuit integrated with a DAC for precise voltage control. The design features N-channel and P-channel MOSFETs ($Q_{1}$, $Q_{2}$), which work in conjunction with differential amplifiers ($A_{1}$, $A_{2}$) and resistive feedback networks ($R_{1}$, $R_{2}$, $R_{3}$) to regulate the output current. The DAC provides a reference voltage ($V_{dac}$), ensuring the circuit maintains stability and accuracy across varying supply voltages ($V_{CC}$, $V_{SS}$). $V_{DD}$ is the reference voltage of the DIDC. This configuration is optimized for low-noise performance, making it ideal for applications requiring high precision and reliability [20].

**Figure 6 sensors-24-06789-f006:**
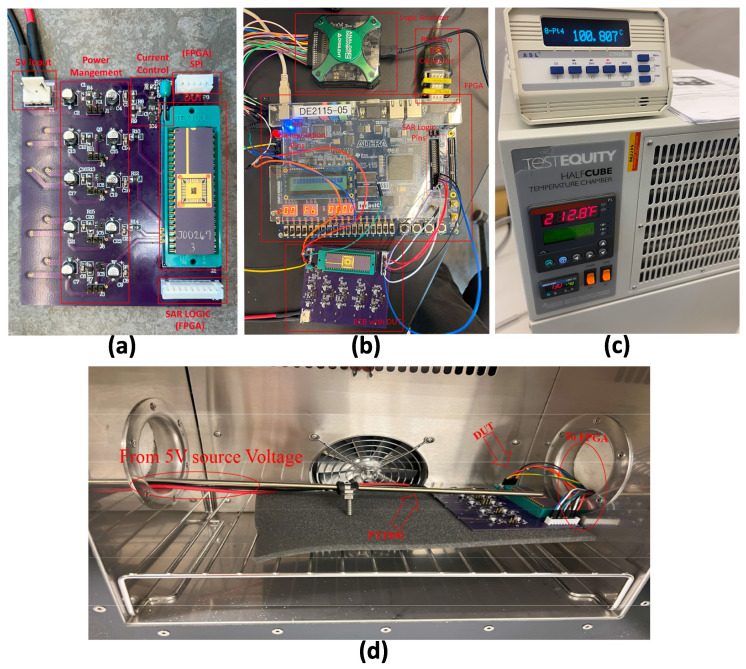
The measurement setups are presented as follows: (**a**) The PCB of the test circuit is shown. As depicted in the photograph, an external 5V power supply is connected to the power lock connector of the circuit. The power management section on the PCB ensures that both the DIDC and the test circuit receive the required supply voltages. The current control circuit from TI is utilized [20]. The DUT is driven by the FPGA through two XH connectors: the top connector drives the SPI, while the bottom connector drives the SAR logic. (**b**) The circuit is tested in the presence of the FPGA. The pins specified for any connections on the FPGA are revealed. An RS232 to USB converter cable delivers data to the computer through UART. A logic analyzer from Analog Devices is used to monitor the SPI communication between the SPI and FPGA. (**c**) A temperature chamber is used to evaluate the operation of the DIDC under hot and cold conditions. A PT1000 sensor is utilized to measure the temperature at the top of the die. (**d**) The setup inside the chamber is shown. The DUT is placed on one side, with the FPGA and external voltage source connected to it. The PT1000 ensures that the temperature on the die remains at the desired level during testing.

**Figure 7 sensors-24-06789-f007:**
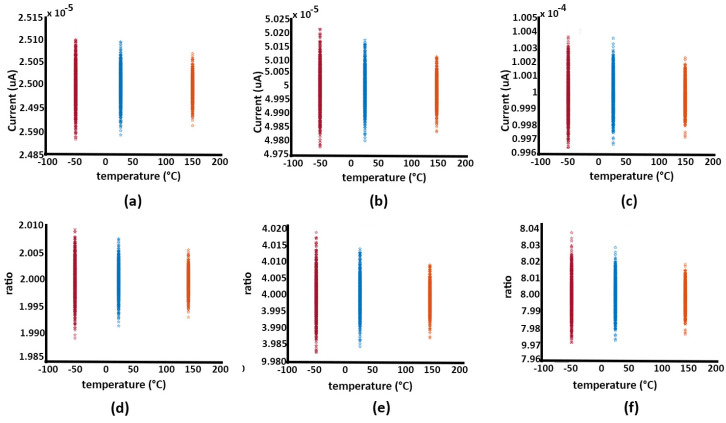
This figure illustrates the current deviations caused by mismatches in the branches of (**a**) bit 5, (**b**) bit 6, and (**c**) bit 7, along with their respective mirroring ratios in (**d**–**f**). To achieve the desired level of accuracy, each branch must maintain its corresponding current within the range defined by the LSB weight.

**Figure 8 sensors-24-06789-f008:**
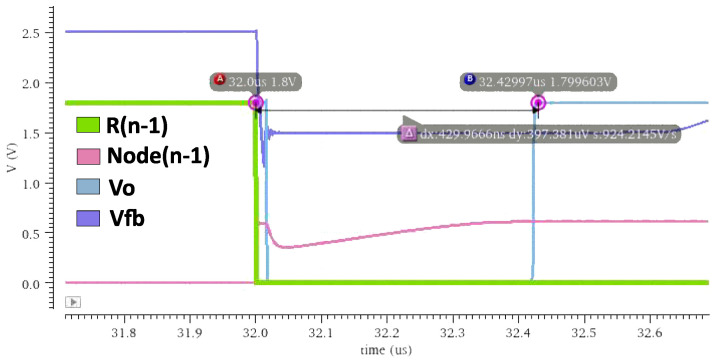
Transient behavior of the nodes involved in the conversion process. From the moment the SAR logic triggers bit n−1 (bit 7), it takes approximately 430 ns for the output to stabilize.

**Figure 9 sensors-24-06789-f009:**
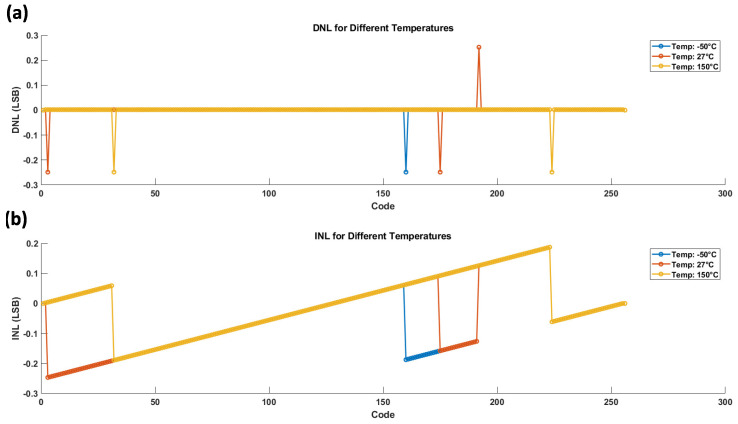
(**a**) DNL and (**b**) INL of the post-layout simulation.

**Figure 10 sensors-24-06789-f010:**
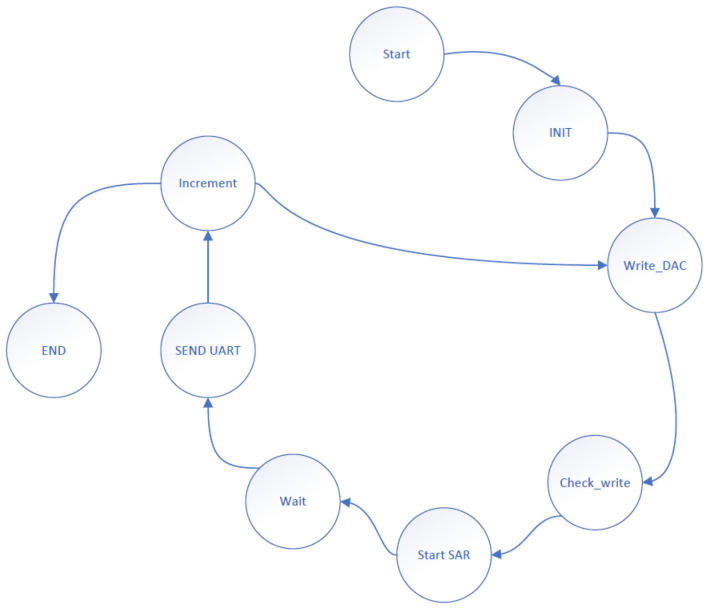
The state machine of the ramp conversion is shown here. The process starts with system initialization in the "Start" and "INIT" states, followed by writing an initial value of zero to the DAC in “Write_DAC”. The “Check_write” state verifies the correct value, and “Start SAR” initiates the SAR logic for conversion. The speed of the conversion is determined here. After waiting in “Wait”, the result is sent via UART (“SEND UART”), and the DAC value is incremented (“Increment”). The cycle repeats until full scale is reached, concluding in the “END” state.

**Figure 11 sensors-24-06789-f011:**
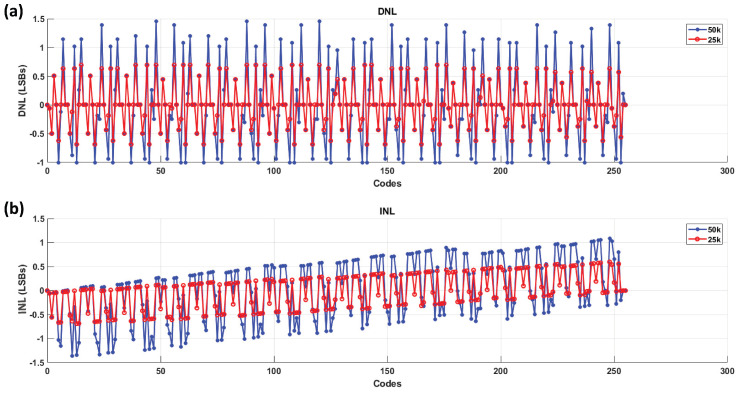
INL and DNL curve is shown here. (**a**) The DNL plots at 25 kHz and 50 kHz SAR clock speeds, respectively, both measured at room temperature. The observed deviation is within ±0.7 LSB at 25 kHz, which is likely attributed to the test bench setup. However, at 50 kHz, missing codes appear, and the deviation increases, indicating reduced accuracy. This limitation arises from the setup’s ability to handle higher speeds. (**b**) The corresponding INL plots at 25 kHz and 50 kHz, respectively.

**Figure 12 sensors-24-06789-f012:**
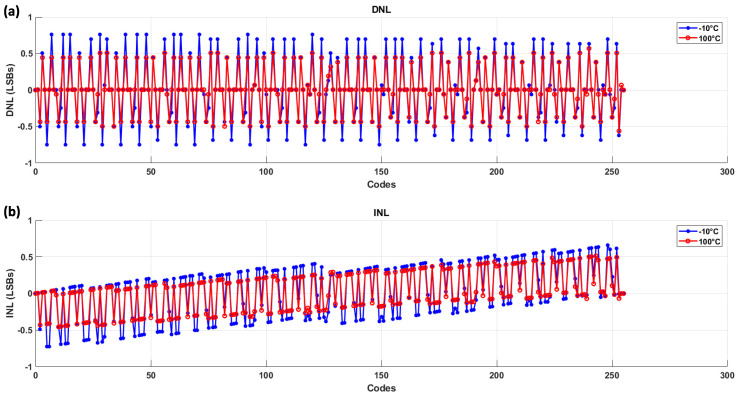
(**a**) DNL plots at −10 °C and 100 °C, measured at 25 kHz SAR clock speed. The results indicate that the DNL deviation remains within ±0.8 LSB for the cold temperature, and less than ±0.5 LSB for the hot temperature. (**b**) INL plots at −10 °C and 100 °C under the same conditions. The INL curve shows a slight increase with code number, but the variation remains within acceptable limits, ensuring that the system maintains its accuracy across these temperature extremes.

**Figure 13 sensors-24-06789-f013:**
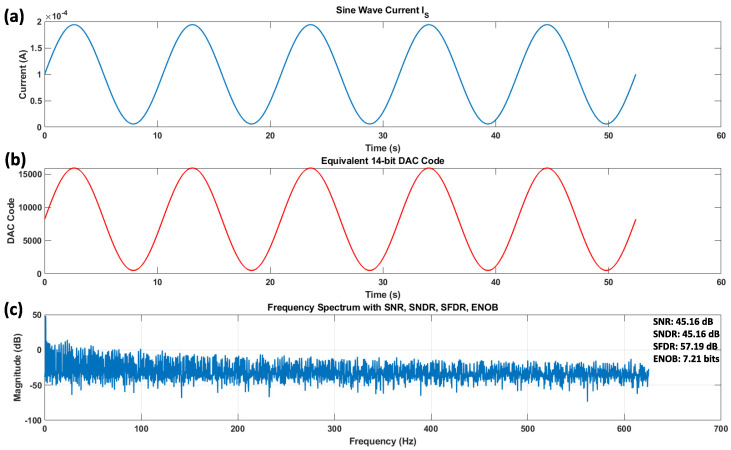
The process of generating DAC code and performing spectrum analysis of the processed output is shown as follows: (**a**) A sine wave representing the $I_{S}$ current, which needs to be fed into the sensor, is depicted here. This sine wave covers a range slightly higher and lower than the minimum and maximum values required for $I_{S}$. (**b**) Using Equation (Equation 11), the corresponding DAC code was generated and programmed into the FPGA, and the resulting Verilog code was saved. (**c**) The spectrum analysis was carried out using the state machine shown in Figure 10, following the procedure outlined from Equations (Equation 23)–(Equation 30).

**Table 1 sensors-24-06789-t001:** Transistor dimensions for different bit levels.

Bit	Type	W (m)	L (m)	Finger	Multiplier
7	base	500 n	16 u	16	8
	cascode	500 n	4 u	16	8
6	base	500 n	16 u	16	4
	cascode	500 n	4 u	16	4
5	base	500 n	16 u	16	2
	cascode	500 n	4 u	16	2
4	base	500 n	16 u	16	1
	cascode	500 n	4 u	16	1
3	base	500 n	16 u	8	1
	cascode	500 n	4 u	8	1
2	base	500 n	16 u	4	1
	cascode	500 n	4 u	4	1
1	base	500 n	16 u	2	1
	cascode	500 n	4 u	2	1
0	base	500 n	16 u	1	1
	cascode	500 n	4 u	1	1

**Table 2 sensors-24-06789-t002:** Current values and standard deviations for each bit at different temperatures ($\times 10^{-4}$).

Temp (°C)		Bit7	Bit6	Bit5	Bit4	Bit3	Bit2	Bit1	Bit0	Dummy
−50	mean	1.00	0.50	0.25	0.13	0.0625	0.0312	0.0156	0.00777	0.00777
	std	0.0138	0.00742	0.00402	0.00229	0.00141	0.000902	0.000611	0.000408	0.000408
27	mean	1.00	0.50	0.25	0.13	0.0625	0.0312	0.0156	0.00778	0.00778
	std	0.0108	0.00579	0.00313	0.00179	0.00111	0.000705	0.000477	0.000319	0.000319
150	mean	1.00	0.50	0.25	0.13	0.0625	0.0312	0.0156	0.00778	0.00778
	std	0.00814	0.00437	0.00235	0.00135	0.000836	0.000532	0.000360	0.000242	0.000242

**Table 3 sensors-24-06789-t003:** Performance summary and comparison with other DxDCs.

Items	This Work	[15]	[16]
Process (nm)	180	65	65
Area (mm^2^)	0.016	1.71	0.007
Power (μW)	117	0.34	2250
Energy/Conv. (pJ)	1.52	N.A.	9 nJ
ENOB	7.21	N.A.	10
Inaccuracy	0.36%	±3.8%	0.75%
Resolution (σ/μ)	0.1375%	2.4%	N.A.
Temperature (°C)	−10 and 100 °C	−20 to +80	−55 to +200

Simulated between −50 and 150 °C.

## Data Availability

The research data can be received through the sending email to the corresponding authors.

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
