# Peer review of "Direct Current to Digital Converter (DIDC): A Current Sensor"

_sensors, 2024, doi:10.3390/s24216789_

Round 1

Reviewer 1 Report

Comments and Suggestions for Authors

The paper is about an integrated circuit designed to measure current without using shunt resistors to convert current-to-voltage.

The circuit was designed and fabricated for a 180 nm silicon node.

In general the paper organization is good. However, the measurement principle is not well explained.

If we consider the following sentence found in the conclusion section: "...in this paper, we introduced a systematic approach to designing a Direct Current-to-Digital Converter (DCDC), exemplified through an 8-bit implementation aimed at advanced current measurement in System-on-Chip (SoC) applications",  we can affirm that the paper does not fullfill its goal, in the sense that i cannot find a methodology design well tailored in the text.

I recommend to give better details regarding the circuits themselves. It is necessary to inform transistor dimesioning details (W and L sizes???).

Please, give more information about the layout dimentions.

Improve figure 3. I cannot find too much useful information from Fig 3.a, for instance.

In the comparison tables, inform the current measurement method used by  references.

A few corrections on the text are also necessary. A few examples below:

- line 126: in Fig ??

- line 134: output current ranging from 100A to 20mA

- separate unities from numbers

Comments on the Quality of English Language

The English language seems ok.

Author Response

Dear Sir/Madam,

Thank you very much for your insightful and constructive feedback on our manuscript. We greatly appreciate your time and effort in reviewing our work. Your valuable suggestions have helped us improve the quality of the paper. We have addressed each of your comments and made the necessary revisions, which are detailed in the responses below.

Please find the revised version of the manuscript and the accompanying code that has been updated accordingly.

Thank you again for your valuable input.

Sincerely,
Saeid karimpour

Reviewer 2 Report

Comments and Suggestions for Authors

This study presents an 8-bit Direct Current-to-Digital Converter (DCDC). The design methodology is promising and the results are presented systematically. The results and discussion methodologically covered the important performance metrics and substantiated the novelty introduced in the design. However, the following comments need to be addressed. 

Introduction

  1. In this context, we propose a novel method that directly measures the current and 63 functions effectively as a current sensor. - The manuscript’s introduction lacks a literature survey of the existing DCDCs. What are the drawbacks of the existing DCDCs? What challenges of DCDCs have been identified and how does the design presented in this study aim to address these challenges? 

Fabrication and Measurement 

  1. Line 136 - This is not an equation. Please delete this equation and merge it into the text.

Circuit Design Method

  1. Line 126 - Figure number missing. Please fix. 

  2. Line 157 - An IS, which is determined to be measured, is generated using a programmable two-stage high-side current source circuit from Texas Instruments - Please mention the part number and the reason for choosing this current source. 

Results

  1. Figure 7 - In this example, as shown in Figure 7, the worst behavior in terms of mismatch 259 occurs at the coldest temperature.- What is the reason for the improvement observed at higher temperatures? This is a key aspect of the study and needs to be elaborated. 

  2. Figure 8 is not referenced anywhere in the study. Please explain Figure 8 in the text. 

  3. Figure 9 is not explained in the text section. What are the inferences from these simulations? Why do the post-layout simulations not show much temperature dependence?

  4. Figure 11 - The y-axis label is supposed to be INL. Please correct this.

Comments on the Quality of English Language

Please do a thorough spell check. 

Author Response

Dear Sir/Madam,

Thank you very much for your insightful and constructive feedback on our manuscript. We greatly appreciate your time and effort in reviewing our work. Your valuable suggestions have helped us improve the quality of the paper. We have addressed each of your comments and made the necessary revisions, which are detailed in the attached below.

Please find the revised version of the manuscript and the accompanying code that has been updated accordingly.

Thank you again for your valuable input.

Sincerely,
Saeid Karimpour

Round 2

Reviewer 1 Report

Comments and Suggestions for Authors

The paper is well revised and can be accepted in this current form.